# Identification of an Oxidosqualene Cyclase Gene Involved in Steroidal Triterpenoid Biosynthesis in *Cordyceps farinosa*

**DOI:** 10.3390/genes12060848

**Published:** 2021-05-31

**Authors:** Gi-Hong An, Jae-Gu Han, Hye-Sung Park, Gi-Ho Sung, Ok-Tae Kim

**Affiliations:** 1Department of Herbal Crop Research, National Institute of Horticultural and Herbal Science, Rural Development Administration, Eumseong 27709, Korea; agiho@korea.kr (G.-H.A.); jaeguhan@gmail.com (J.-G.H.); hyesung2@korea.kr (H.-S.P.); 2Catholic Kwandong University International St. Mary’s Hospital, Incheon 22711, Korea; sung97330@gmail.com

**Keywords:** *Cordyceps farinosa*, fusidic acid, oxidosqualene cyclase, protostadienol synthase, triterpenoid

## Abstract

Various fungi including *Cordyceps farinosa*, an entomopathogenic fungus, can produce steroidal triterpenoids. Protostadienol (protosta-17(20)Z,24-dien-3β-ol) is a precursor of steroidal triterpenoid compounds. To identify oxidosqualene cyclase (OSC) gene candidates involved in triterpenoid biosynthesis, genome mining was performed using Illumina sequencing platform. In the sequence database, two OSC genes, *CfaOSC1* and *CfaOSC2*, in the genome of *C. farinosa* were identified. Predicted amino-acid sequences of CfaOSC2 shared 66% similarities with protostadienol synthase (OSPC) of *Aspergillus fumigatus*. Phylogenetic analysis showed a clear grouping of CfaOSC2 in the OSPC clade. Function of CfaOSC2 was examined using a yeast INVSc1 heterologous expression system to endogenously synthesize 2,3-oxidosqualene. GC–MS analysis indicated that CfaOSC2 produced protosta-13(17),24-dien-3β-ol and protostadienol at a 5:95 ratio. Our results demonstrate that CfaOSC2 is a multifunctional triterpene synthase yielding a predominant protostadienol together with a minor triterpenoid. These results will facilitate a greater understanding of biosynthetic mechanisms underlying steroidal triterpenoid biosynthesis in *C. farinosa* and other fungi.

## 1. Introduction

Steroidal triterpenoids such as ganoderic acid, pachymic acid, fusidic acid, and helvolic acid are produced by fungal cells. These compounds can protect fungal cells against other pathogens. Ganoderic acid and pachymic acid have anti-cancer, immune-enhancing, and anti-inflammation effects [1,2,3]. Fusidic acid and helvolic acid are widely used as antibiotics against Gram-positive bacteria. The biosynthesis mechanism of fusidic acid and helvolic acid has been recently elucidated using a heterologous expression system [4,5]. However, the pathway of other steroidal triterpenoid biosynthesis is not clear yet.

A number of studies have suggested that lanosterol and protostadienol (protosta-17(20)Z,24-dien-3β-ol) are precursors of steroidal triterpenoid compounds [6,7]. In the case of triterpene biosynthesis in fungal cells, lanosterol synthase (OSLC) and protostadienol synthase (OSPC) belonging to a class of oxisqualene cyclases (OSCs) are involved in the biosynthesis of lanostane-type and fusidane-type triterpenoid, respectively (Figure 1). It has been reported that these OSCs are key enzymes that regulate biosynthesis [8]. In both fungi and animals, OSLC is responsible for producing sterol precursors for cholesterol and ergosterol. OSLC genes from several fungal species, animals, plants, and microorganisms have been characterized [9,10,11,12,13]. Interestingly, genomes of some fungi encode more than one OSC. Other OSCs can also produce triterpenoid secondary metabolites via their OSC function. So far, only one OSPC gene has been characterized in *A. fumigatus* through a heterologous expression system in yeast cells [6]. OSPC gene from other organisms has not been characterized yet.

The species *Cordyceps farinosa* belongs to the Cordycipitaceae family in the Hypocreales order of Ascomycota and is well-known as entomopathogenic fungi with a worldwide distribution in temporate and tropical zones [14]. These entomopathogenic fungi can also produce a variety of chemicals that cause severe adverse reactions in other organisms, including bacteria and other animals. *Sarocladium oryzae*, *Acremonium fusidioides*, *Metarhizium anisopliae*, and *Aspergillus fumigatus* are fungi that can produce fusidanes as steroidal antibiotics [15]. In particular, *Cordyceps kogane* can produce fusidic acid that is structurally similar to fusidane [16].

To discover genes related to triterpenoid biosynthesis, *Cordyceps farinosa* mycelium was chosen as a biosynthesis model of steroidal triterpenoid. We chose *C. farinosa* because, in the preliminary comparative genome analysis of Hypocrealean species, we recognized that a gene cluster of *C. farinosa* was similar to a well-known gene cluster related to helvolic acid biosynthesis in *A. fumigatus*. In the present study, whole genome sequences of *C. farinosa* KMCC47486 were analyzed and two OSC genes, lanosterol synthase and protostadienol synthase, were found. We isolated a gene encoding a putative protostadienol synthase from *C. farinosa* mycelium. Its function was then determined using a heterologous expression system in yeast along with product analysis using GC-MS. In addition, a gene cluster including OSPC gene was identified. It consisted of seven genes highly conserved in *A. fumigauts* for helvolic acid biosynthesis. We also discussed triterpenoids produced by discovering genes that matched with the gene cluster related to helvolic acid biosynthesis.

## 2. Materials and Methods

### 2.1. Strain of C. farinosa

A strain of *C. farinosa* (KACC47486) was obtained from Korean Agricultural Culture Collection (KACC), Rural Development Administration (RDA), Republic of Korea. The mycelium of *C. farinosa* was cultured on PDA (Potato Dextrose Agar) medium at 25 °C in darkness for 2 weeks. To identify the strain through sequence comparison of its ITS rDNA region, phylogenetic tree analysis was performed using the Neighbor-Joining method.

### 2.2. Genome Sequencing and Assembly

Genomic DNA was extracted from freeze-dried mycelium using a DNeasy Plant Mini Kit (Qiagen, Valencia, CA, USA) according to the manufacturer’s instructions. Concentration and purity of the extracted genomic DNA were assessed using a NanoDrop ND-1000 spectrophotometer (Thermo–Fisher Scientific, Waltham, MA, USA). Next generation sequencing (NGS) library construction and sequencing were carried out by Macrogen Inc. (Seoul, Korea). Paired-end and mate-pair libraries with insert sizes of 300 bp and 5 kbp, respectively, were prepared. Subsequently, NGS was carried out using an Illumina HiSeq platform (KAPA Biosystems, Roche, San Francisco, CA, USA) with a coverage of approximately 350×. Generated sequence reads with low-quality scores or short lengths were filtered out using a FASTX Toolkit 0.0.13 (http://hannonlab.cshl.edu/fastx_toolkit (accessed on 28 May 2021)). Trimmed paired-end and mate-pair reads were then assembled using a Velvet 1.2.01 assembler [17]. All reads were deposited at NCBI, and can be accessed in the BioProject data base under project accession number PRJNA314175.

### 2.3. Phylogenetic Analysis

Amino acid sequences of OSCs in *C. farinosa* and its related species were extracted using tblastn of NCBI BLAST 2.2.28+ package [18] with a query sequence of *A. fumigatus* OSLC (XP_747936) against selected fungal genomes belonging to the order Hypocreales: *Beauveria bassiana* ARSEF2860 (PRJNA38719), *Claviceps purpurea* 20.1 (PRJEA76493), *Cordyceps militaris* CM01 (PRJNA41129), *Epichloë festucae* FI1 (PRJNA51625), *Epichloë typhina* E8 (PRJNA174036), *Fusarium graminearum* PH-1 (PRJNA243), *Fusarium oxysporum* Fo5176 (PRJNA68027), *Fusarium verticillioides* 7600 (PRJNA15553), *Metarhizium acridum* CQMa 102 (PRJNA38715), *Metarhizium anisopliae* ARSEF23 (PRJNA38717), *Tolypocladium inflatum* NRRL8044 (PRJNA73163), *Trichoderma atroviride* IMI206040 (PRJNA19867), *Trichoderma reesei* QM6a (PRJNA15571), *Trichoderma virens* Gv29-8 (PRJNA19983), *Beauveria pseudobassiana* KACC47484 (PRJNA314175), *Beauveria sungii* KACC47481 (PRJNA314175), *Cordyceps pruinosa* KACC44470 (PRJNA314175), *Cordyceps farinosa* KACC47486 (PRJNA314175), and *Cordyceps tenuipes* KACC47485 (PRJNA314175). Predicted amino acid sequences were aligned using Clustal Omega [19] with default settings and then manually adjusted. Phylogenies were inferred using neighbor joining (NJ) and maximum likelihood (ML) methods of a MEGA6 software [20]. Relative robustness of individual branches was estimated with 500 replicates using bootstrapping (BS).

### 2.4. Generation of Plasmid Vectors

Sequences for two *C. farinosa* genes were deposited in GenBank with accession numbers of MF972281 for CfaOSC1 and MF972287 for CfaOSC2. To amplify the CfaOSC2 gene, forward (5′-TGATGCCTGTCGCCGATATTGAC-3′) and reverse (5′-TTATCCTTTGTAAAAGGACGCTC-3′) primers were designed. PCR was then performed with an Ex-taq DNA polymerase (Takara, Tokyo, Japan) using the following cycling conditions: 30 cycles at 95 °C for 1 min, 58 °C for 1 min, and 72 °C for 2 min; followed by a final extension step at 72 °C for 10 min. The obtained fragment was purified and cloned into a pYES2.1 vector using a TOPO TA-expression kit (Invitrogen, Carlsbad, CA, USA), enabling the construction of an expression plasmid pYES-CfaOSC2 in yeast cells. After sequence (pYES-CfaOSC2) confirmation, the plasmid was used for yeast transformation.

### 2.5. Functional Expression of CfaOSC2 in Yeast

Functional characterization was carried out in yeast strain INVSc1 purchased from Invitrogen (Carlsbad, CA, USA). Yeast transformation and insert-DNA overexpression were carried out as described by Kushiro et al. [21]. Single clones including pYES–CfaOSC2 were incubated in 15 mL of a synthetic complete medium containing 2% glucose without uracil at 30 °C with shaking at 220 rpm for two days. After three-day induction with 2% galactose, cells from two 50 mL conical tubes were collected into one tube, refluxed with 8 mL of 20% potassium hydroxide (KOH) and 50% ethanol (EtOH), and extracted three times with hexane at the same volume. Extracts were concentrated under a stream of nitrogen gas (N_2_) and resuspended in 1 mL of chloroform (CHCl_3_). Subsequently, 100 µL of the extract was transferred into a new vial and concentrated under a stream of N_2_ at 70 °C. For GC–MS analysis, the extract was silylated with 50 µL of *N*,*O*-bis(trimethylsilyl)trifluoroacetamide (BSTFA) and 50 µL of pyridine for 30 min at 70 °C.

### 2.6. GC-MS Analysis

GC–MS analyses were run under the same conditions as described by Wang et al. [22]. Briefly, a 1-μL aliquot of the solution was analyzed using a 7890N gas chromatography (Agilent Technologies, Santa Clara, CA, USA) equipped with a 5973-inert mass spectrometer (MS) detector (Agilent Technologies, Santa Clara, CA, USA) and an Agilent HP-5 capillary column (length of 30 m, i.d. of 250 μm, film thickness of 0.25 μm). The injection temperature was set at 50 °C for 2 min. The column-temperature program was as follows: 40 °C/min ramp to 200 °C, a hold at 200 °C for 2 min, followed by an increase to 320 °C at a rate of 3 °C/min; and lastly a hold at 320 °C for 30 min. Triterpenoids were identified by comparison with authentic extracts from yeast cells co-expressing AfuOSC3 (*A. fumigatus* protostadienol synthase) and CYP5081A1 (*A. fumigatus* cytochrome P450 oxidase) as reported by Mitsuguchi et al. [7]. The constructed vector of pESC(-Ura)-AfuOSC3/CYP5081A1 was kindly provided by Dr. Tetsuo Kushiro (Meiji University, Tokyo, Japan). As previously described, the same protocol was applied for the transformation and culturing of yeast strain INVSc1 with the vector.

### 2.7. Synteny Analysis

The OSC gene cluster of *A. fumigatus* Af293 proposed by Lodeiro et al. [6] was retrieved from GenBank (XP_751348-XP_751356), including a series of genes catalyzing monooxygenation, dehydrogenation, and acyl transfer to convert protostadienol into helvolic acid. Based on the result of our phylogenetic analysis, the synteny of OSC gene cluster of *A. fumigatus* was searched within the three selected hypocrealean species., viz. *M. anisopliae*, *B. sungii*, and *C. farinosa* (Group III in Figure 4). Orthologous regions were identified using the reciprocal best hit (RBH) approach [23]. First, local databases were created from genome sequences of *C. farinosa*. After BLAST-searching *AfuOSC3* gene clusters in databases, nucleotide sequences highly matched with protein sequences of AfuOSC3 were extracted from each genome. They were then conversely queried against the *AfuOSC3* gene cluster. In each pair-wise comparison, reciprocally best-matched genes (or regions) were regarded as orthologs.

## 3. Results and Discussion

### 3.1. Identification of C. farinosa

The mycelium of putative *C. farinosa*, which was provdied by the KACC (Korean Agriculture Culture Collection), was cultured on PDA medium for 2 weeks (Figure 2a). To identify the strain, the fungal source was analyzed by ITS gene sequencing. The BLAST search of the ITS sequence of the strain showed the highest similarity to *Cordyceps* and the phylogenetic tree was constructed using MEGA 5.2 software. As shown in Figure 2b, this strain clearly was grouped with *C. farinosa*. In addition, *Isaria farinosa* as the scientific name has long been used for this fungus. However, the genetic position of *Isaria* was changed to the *Cordyceps* by the principal of priority because of the discontinuance of dual nomenclature for pleomorphic fungi in 2011 [24]. After confirmation of the strain, we performed genome sequencing.

### 3.2. Sequence Features of Two OSC Genes

Mass DNA-sequence data were produced using an Illumina sequencing platform to identify OSC genes involved in triterpenoid biosynthesis. In the sequence database, two *OSC* genes, *CfaOSC1* (*C. farinosa* lanosterol synthase) and *CfaOSC2* (*C. farinosa* protostadienol synthase), were identified from the genome of *C. farinosa*. Both genes contained full-length cDNAs. Open reading frames (ORFs) of *CfaOSC1* and *CfaOSC2* were 2223 bp and 2205 bp in length, respectively. These ORFs were predicted to encode CfaOSC1 and CfaOSC2 proteins having 741 and 735 amino acids with masses of 84.407 kDa and 82.615 kDa, respectively. Deduced amino acid sequences of CfaOSC1 (70%) and CfaOSC2 (66%) were orthologous to lanosterol synthase (Afu5g04080) and protostadienol synthase (AfuOSC3) of *A. fumigatus*, respectively. As shown in Figure 3, CfaOSC1 and CfaOSC2 contained repeats of the QW motif [25], a typical feature of the triterpene-synthase superfamily. CfaOSC2 consisted of ^699^ACPGGMR^705^ motif in the C-terminal region. This motif is known to play a role in protostadienol formation [26].

To determine the relationship of these two CfaOSCs with other fungal OSCs, phylogenetic analysis was performed as to determine its relationships with other fungi (Figure 4). As expected, CfaOSC1 and CfaOSC2 were clearly grouped in OSLC and OSPC clades, respectively. Therefore, CfaOSC2 is likely to have function in protostadienol production.

### 3.3. Functional Characterization of CfaOSC2 in Yeast

Full-length cDNA of *CfaOSC2* was amplified using polymerase chain reaction (PCR) to elucidate the function of *CfaOSC2* gene in steroidal triterpenoid biosynthesis. The full-length cDNA was cloned into a yeast expression vector pYES2.1 under the control of a GAL1 promotor. The *CfaOSC2* gene functionally was characterized using a heterologous expression system in yeast INVSc1 that could endogenously synthesize 2,3-oxidosqualene. After three days of galactose induction, yeast cells were harvested and then extracted with hexane. To check whether or not alcohol triterpenes were extracted from yeast cells, we added lupeol as a internal standard before the extraction. Results of gas chromatography–mass spectrometry (GC–MS) analysis indicated that CfaOSC2-overexpressing yeast cells produced two products not present in control yeast cells carrying an empty vector pYES2 (Figure 5a). Peaks of yeast cells co-expressing AfuOSC3/CYP5081A1 as described by Mitsuguchi et al. [7] were used as authentic standards. Results of retention-time comparison indicated that these two products at a ratio of 5:95 were protosta-13(17),24-dien-3β-ol (1) and protosta-17(20)Z,24-dien-3β-ol (2, protostadienol), respectively. In addition, the pattern of mass fragments of these identified compounds in yeast cells overexpressing AfuOSC3/CYP5081A1 was compared. Results are shown in Figure 5b. Both peaks were alcohol triterpenes as they comprised the *m*/*z* 498 molecular ion which was trimethylsilylated (TMS). In addition, the characteristic fragment ions in MS spectrum of protostadienol were *m*/*z* = 191, 339 and 429, and these values were consistent with those reported by Ledro et al. [6] (Appendix A). Therefore, CfaOSC2 is a multifunctional triterpene synthase producing compounds **1** and **2**.

### 3.4. Synteny Comparison of a Gene Cluster for Helvolic Acid Biosynthesis

Whole fungi-genome sequences are available from public databases. Lodeiro et al. [6] and Mitsuguchi et al. [7] have identified a gene cluster related to helvolic acid biosynthesis that consists of OSPC, four cytochrome P450 (CYP) monooxygenases, two transferase-family proteins, and two dehydrogenase genes. Nine of these gene-cluster genes have been recently characterized using a heterologous-expression system in *Aspergillus oryzae* NSAR1, resulting in the detection of helvolic acid together with 21 protostadienol derivatives [4]. We compared the gene cluster of *A. fumigatus* for helvolic acid biosynthesis with that of *C. farinosa* based on DNA sequences (Figure 6). A gene cluster of *C. farinosa* consists of seven genes, encoding one protostadienol synthase, three cytochrome P450 (CYP), one acyltransferase, one short-chain dehydrogenase/reductase and one ketosteroid dehydrogenase. Among seven genes, amino acids of five genes are highly similar to those of HelA (65%), HelB1 (64%), HelB2 (60%), HelC (61%) and HelB4 (66%). However, amino acids of two genes encoding acyltransferase and ketosteroid dehydrogenase showed the low level of similarity to those of HelD2 (38%) and HelE (26%). The gene cluster of *C. farinosa* showed deletion of a CYP gene (Gene No. 5) and an acyltransferase gene (Gene No. 6) compared to the helvolic acid cluster. We performed HPLC analysis to determine helvolic acid in *C. farinosa* (Appendix A). The result indicated that a peak which was not consistent with that of helvolic acid was detected. Therefore, the result might indicate that helvolic acid is not biosynthesized because of deletion of the two genes in *C. farinosa*. Based on these results, we depicted a putative pathway of triterpenoid biosynthesis in *C. farinosa* (Figure 7). We inferred that the final product of operating this gene cluster might be 16β-acetyloxy-29-norprotosta-1,17(20)Z,24-trien-3-one-21-oic acid considering that those two genes (Gene No. 5 and Gene No. 6) were deleted in the gene cluster. Future studies are needed to identify all genes except *OSPC* gene using a heterologous expression system.

## 4. Conclusions

This study revealed that the *CfaOSC2* gene encoded an enzyme that could catalyze the transformation of oxidosqualene to a predominant protostadienol along with a minor triterpenoid and that *CfaOSC2* was associated with protostane-type triterpenoid biosynthesis using a heterologous expression system in yeast cells. However, the next catalytic step of the cycling triterpenoid remains unclear. Based on our DNA-sequence database, future studies need to elucidate the function of candidate genes in the gene cluster associated with *CfaOSC2*. In addtion, it will be necessary to identify 16β-acetyloxy-29-norprotosta-1,17(20)Z,24-trien-3-one-21-oic acid or any triterpene compound in *C. farinosa* cells. Results of this study provide a greater understanding about biosynthetic mechanisms underlying steroidal triterpenoid biosynthesis in *C. farinosa* and other fungi.

## Figures and Tables

**Figure 1 genes-12-00848-f001:**
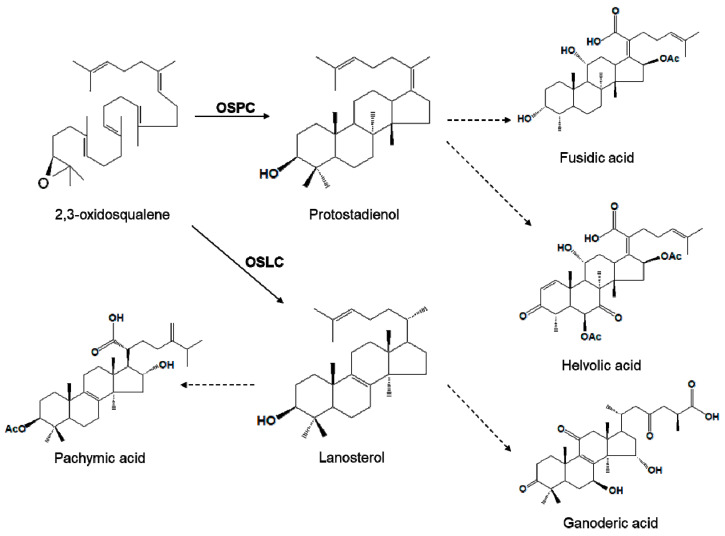
Putative biosynthesis pathway of steroidal triterpenoids in fungi. Dot arrows mean multiple steps.

**Figure 2 genes-12-00848-f002:**
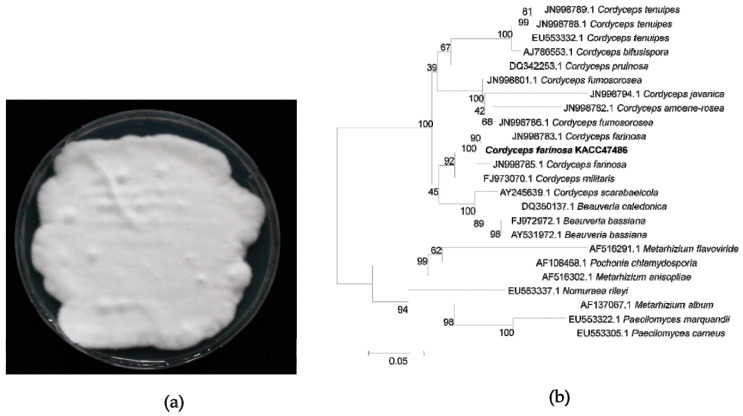
Identification of *Cordyceps farinosa* (KACC47486). (**a**) Strain of *C. farinosa* (KACC47486) cultured on PDA medium for two weeks; (**b**) Phylogenetic analysis using Neighbor-Joining method based on partial sequences of the ITS rDNA region.

**Figure 3 genes-12-00848-f003:**
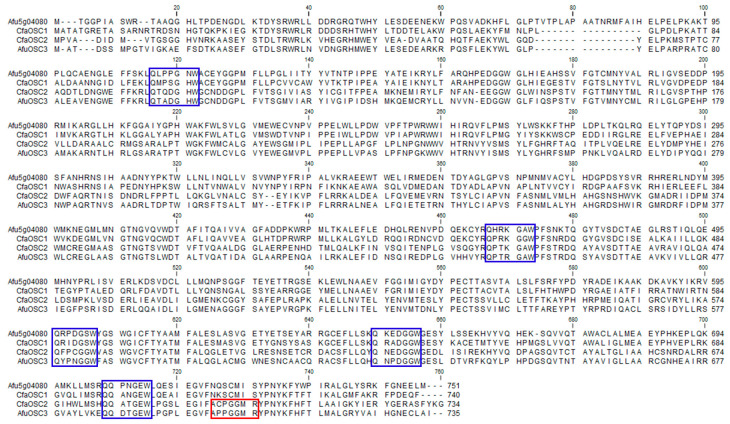
Alignment of amino-acid sequences of OSC genes in *Cordyceps farinosa* and *Aspergillus fumigatus*. Blue and red boxes indicate the QW(QXXXGXW) motif and the specific motif that plays a role in protostadienol production, respectively.

**Figure 4 genes-12-00848-f004:**
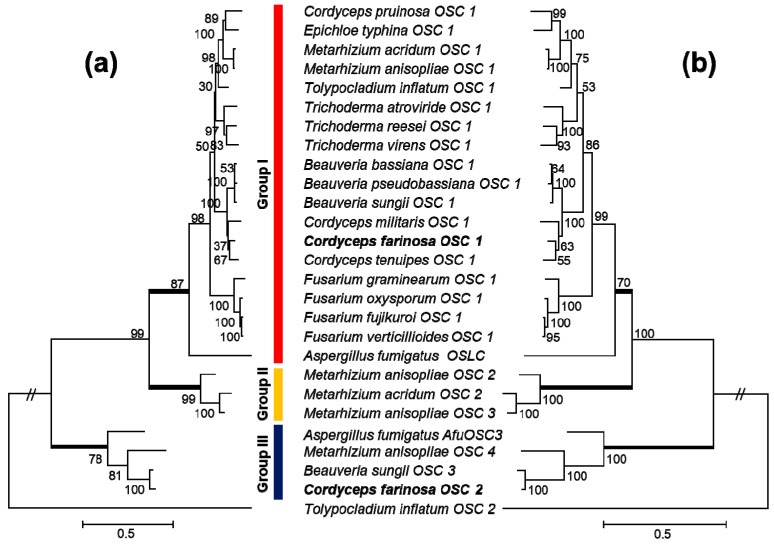
Phylogenetic tree of oxidosqualene cyclase (OSC) sequences extracted from selected Hypocreales species and inferred by neighbor joining (NJ, **a**) and maximum likelihood (ML, **b**) analyses. The number of changes among sequences is presented by branch length. The scale bar equals the number of nucleotide substitutions per site. Group I comprises genes homologous to lanosterol synthase (OSLC). Group III delimits orthologous of protostadienol synthase (OSPC). Three uncharacterized OSCs from *Metarhizium* spp. are nested in Group II.

**Figure 5 genes-12-00848-f005:**
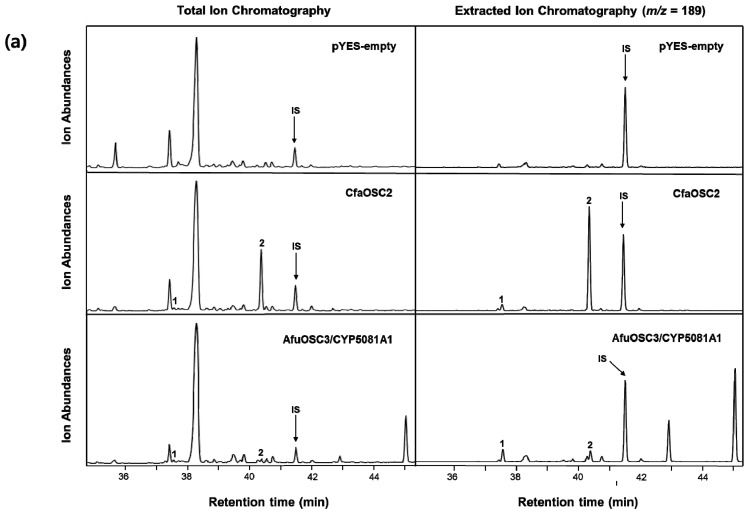
Identification of triterpenoids in CfaOSC2-overexpressing yeast cells by gas chromatography–mass spectrometry (GC–MS) analysis. (**a**) GC chromatograms from the strain expressing the *CfaOSC2* gene, the control yeast strain harboring the pYES2 vector, and the authentic standards from the yeast strain co-expressing *AfuOSC3*/*CYP5081A1* genes. IS means internal standard (Lupeol). (**b**) MS spectra of trimethylsilylated triterpenoids produced from the CfaOSC2-expressing strain (left) and the standard (right).

**Figure 6 genes-12-00848-f006:**
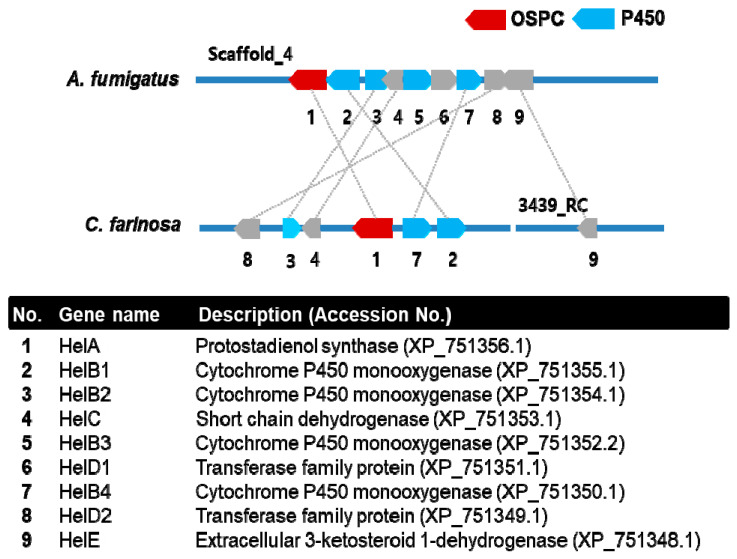
Synteny comparison of two gene clusters involved in triterpenoid biosynthesis.

**Figure 7 genes-12-00848-f007:**
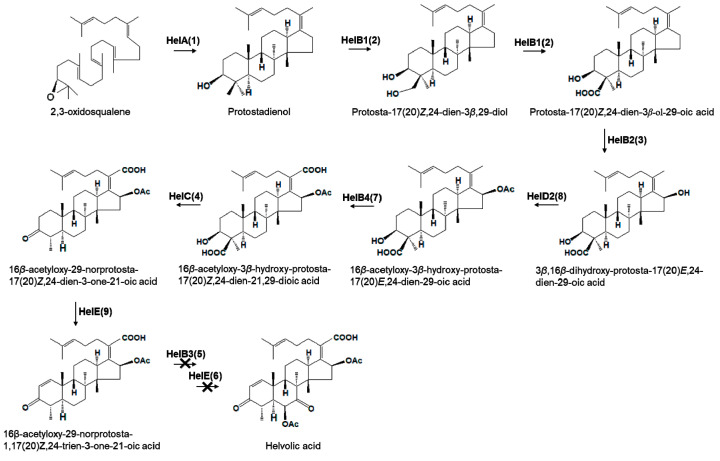
Putative pathways of triterpenoid biosynthesis in *C. farinosa*.

## Data Availability

The data presented in this study are available on request from the corresponding author.

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
