# Peer review of "Identification of an Oxidosqualene Cyclase Gene Involved in Steroidal Triterpenoid Biosynthesis in Cordyceps farinosa"

_genes, 2021, doi:10.3390/genes12060848_

Round 1

Reviewer 1 Report

In the manuscript titled “Identification of an oxidosqualene cyclase gene involved in 2 steroidal triterpenoid biosynthesis in Cordyceps farinosa” by An et al authors identified two oxidosqualene cyclase (OSC) genes CfaOSC1 and CfaOSC2 using whole genome sequencing and phylogenetic analysis. The computational analysis was complemented with heterologous expression and functional characterization of CfaOSC2 using GC-MS analysis followed by synteny analysis with previously characterized gene cluster of A. fumigatus for helvolic acid biosynthesis.

This original work embodies a perfect balance between bioinformatics analysis and heterologous expression for functional characterization and elucidation of novel biosynthetic pathways for steroidal triterpenoid biosynthesis. The methods are well described for easy replication and reproducibility of the experiments performed. Yet, there are some issues that need to be resolved. The following comments might be addressed while preparing a revised version of their article.

Broad comments 

While the authors have mentioned in great detail the importance of the oxidosqualene cyclase genes, steroidal triterpenoids and their biosynthesis, the significance of choosing C. farinosa as the biosynthesis model is not clear. Authors might want to state the significance of these fungus over other Hypocreales or other fungus.

Further, quite recently this fungus (along with many more) has been designated to Cordyceps clade by Kepler et al., 2017 (https://doi.org/10.5598/imafungus.2017.08.02.08). It was hard to review existing literature for C. farniosa as well as C. koganes other than the one authors have referred to (reference 16). The authors need to mention the synonyms and relevant references that exist for these fungi for wider readability of this article.

The bioinformatics tools used in this work are latest and relevant along with the analysis performed. Authors have used public databases and tools such as NCBI BLAST and GenBank for the analysis. There are few things in these sections that need more details/clarification for better understanding the design of this study. Firstly, what was the selection criteria for choosing the fungal genomes for tblastn followed by phylogenetical analysis (Figure 4)? Needs to be mentioned in section 2.3 in methods section.

Authors mention in Section 2.7, Synteny analysis, “best hit sequences were extracted from each genome”. How many genomes were analyzed and what is the criteria for a “best hit”. What does 65% identity translate to for each of the genes in the cluster? What tool was used for Synteny analysis?

In Section 3.2 it is mentioned that using GC–MS analysis CfaOSC2-overexpressing yeast cells produced two products (Peak 1 and 2) that are not present in control yeast cells carrying an empty vector pYES2 as shown in Figure 5a. It seems there is a peak in pYES-empty chromatograms at the same retention time as where peak 1 comes up. Please comment. What was the internal standard used?

Also, has the genomic sequencing data generated in this work uploaded to open access database (NCBI, SRA, etc.)?

Authors might want to discuss more on how the results generated in this study provide a better understanding of regulatory mechanisms underlying steroidal triterpenoid biosynthesis in C. farinosa beyond the proposed pathway (Figure 7) of triterpenoid biosynthesis.

Specific comments 

  • Line 129-130 page 4 – How was the sequence for pYES-CfaOSC2 confirmed?
  • Mention gene details when the gene is first mentioned in main text (e.g. AfuOSC3 in line)
  • Spell check throughout text (e.g. line 68 – A. fumigatus, figure 2 legend – KACC4786 or KACC47486?)

Author Response

Thank you for valuable comments. We did our best to revise this manuscript according to reviewer's comment. Revised parts in this manuscript were remarked by red color.

Once again, thank you for comments.

Reviewer 1

While the authors have mentioned in great detail the importance of the oxidosqualene cyclase genes, steroidal triterpenoids and their biosynthesis, the significance of choosing C. farinosaas the biosynthesis model is not clear. Authors might want to state the significance of these fungus over other Hypocreales or other fungus.

> ANSWER : Thank you for this comment. Please see line 64-67. C. farinosa is an insect-pathogenic fungus, well-known for producing a number of secondary metabolites. This fungus is easy to handle in laboratory since it grows fast on artificial medium. Before starting this work, the authors preliminarily searched genes that tentatively involve in synthesizing oxidosqualene cyclases in species assigned to the order Hypocreales. As we found that C. farinosa was the only species having the gene cluster related to helvolic acid biosynthesis in A. fumigatus, C. farinosa was choosen as the model for the present study. Please see line 64-67.

Further, quite recently this fungus (along with many more) has been designated to Cordyceps clade by Kepler et al., 2017 (https://doi.org/10.5598/imafungus.2017.08.02.08). It was hard to review existing literature for C. farniosa as well as C. koganes other than the one authors have referred to (reference 16). The authors need to mention the synonyms and relevant references that exist for these fungi for wider readability of this article.

> ANSWER : Thank you for this comment. Please see line 182-185. We added the reference.

The bioinformatics tools used in this work are latest and relevant along with the analysis performed. Authors have used public databases and tools such as NCBI BLAST and GenBank for the analysis. There are few things in these sections that need more details/clarification for better understanding the design of this study. Firstly, what was the selection criteria for choosing the fungal genomes for tblastn followed by phylogenetical analysis (Figure 4)? Needs to be mentioned in section 2.3 in methods section.

> ANSWER : Thank you for this comment. Since there is sequence information for AfuOSC3 gene, first of all, we tried randomly to find out OSPC from other fungi and we recongized that a few species had protostadienol synthase in the order Hypocreales and then, in only Hypolcreales, OSC gene sequences from possible species having genome database were extracted and created the phylogenetic tree. 

Authors mention in Section 2.7, Synteny analysis, “best hit sequences were extracted from each genome”. How many genomes were analyzed and what is the criteria for a “best hit”. What does 65% identity translate to for each of the genes in the cluster? What tool was used for Synteny analysis?

> ANSWER : Thank you for this comment. Please see line 172 for “best hit”. Please see line 267-273 for identity of each gene. we added information of genes in the gene cluster, in detail. And for synteny analysis we did use best match for each gene as described in Materials and methods, because it possible to find each gene in gene clusters and number of samples is only three. 

In Section 3.2 it is mentioned that using GC–MS analysis CfaOSC2-overexpressing yeast cells produced two products (Peak 1 and 2) that are not present in control yeast cells carrying an empty vector pYES2 as shown in Figure 5a. It seems there is a peak in pYES-empty chromatograms at the same retention time as where peak 1 comes up. Please comment. What was the internal standard used?

> ANSWER : That's right. When we also saw the TIC, we could not distinguish the peaks between the control and CfaOSC2 overexpression because of very small peak of 1 in CfaOSC2 TIC. So we extracted a chromatography of extracted ion (m/z=189) and you can see very close to a small peak on peak 1 in EIC of CfaOSC2. If you magnify the figure, you easily can see the difference. To confrim whether or not alcohol compounds are extracted from yeast cells, we used lupeol having similar structure to protostadienol as internal standard.

Also, has the genomic sequencing data generated in this work uploaded to open access database (NCBI, SRA, etc.)?

> ANSWER : Thank you for this comment. Please see line 97-98. We added the accession number of BioProject on NCBI database.

Authors might want to discuss more on how the results generated in this study provide a better understanding of regulatory mechanisms underlying steroidal triterpenoid biosynthesis inC. farinosabeyond the proposed pathway (Figure 7) of triterpenoid biosynthesis.

Specific comments

Line 129-130 page 4 – How was the sequence for pYES-CfaOSC2 confirmed?

> ANSWER : Thank you for this comment. Generllay, we did sequence the cloned gene by using primer which was designed from sites in pYES2 vector before expression experiment.

Mention gene details when the gene is first mentioned in main text (e.g. AfuOSC3 in line)

Spell check throughout text (e.g. line 68 – A. fumigatus, figure 2 legend – KACC4786 or KACC47486?)

> ANSWER : Thank you for this comment. We have revised it. Please see lin 157 and 192.

Reviewer 2 Report

This is an interesting piece of work that clearly identifies the role of a novel gene in a secondary metabolic pathway of interest.  It has been performed very well.

In the introduction it would be nice to say something about Cordyceps farinosa - what insects it infects, where in the world it is found, why it was selected, is it easy to work with, is much known about it, are there doubts about its taxonomy (and hence the ITS sequencing and dendrogram?) ..... Just a few sentences?  Also, clarify which organisms should be mentioned in line 63 homologous or orthologous, not associated (unless authors are proposing horizontal gene transfer from A. fumigatus?) and at line 68 Shouldn't the organism be Cordyceps farinosa, not A. fumigatus?

Very minor editing of grammar/english is needed. e. g. line 32 delete the word other; line 41 should be in the case line 46 should be fungal not fungi; line 49/50 do the authors mean that only one OSPC gene has been characterised in Aspergillus fumigatus, or that this is the only OSPC gene that has been characterised so far (probably the latter since the following sentence essentially says this)? line 55 several (or many) rather than different lines 236-238 Can this sentence be written more clearly? line 266 an, not a

Author Response

Thank you for valuable comments. We did our best to revise this manuscript according to reviewer's comment. Revised parts were remarked by red color. 

Once again, thank you for comments.

Reviwer 2

This is an interesting piece of work that clearly identifies the role of a novel gene in a secondary metabolic pathway of interest.  It has been performed very well.

In the introduction it would be nice to say something about Cordyceps farinosa - what insects it infects, where in the world it is found, why it was selected, is it easy to work with, is much known about it, are there doubts about its taxonomy (and hence the ITS sequencing and dendrogram?) ..... Just a few sentences? 

> ANSWER : We revised C. farinosa information. Please see line 55-57 and according to your opinion plus Reviwer 3's opinion, the part of identification of a strain used in this work shifted to Results and Discussion section. Please see line 177-187.

Also, clarify which organisms should be mentioned in line 63 homologous or orthologous, not associated (unless authors are proposing horizontal gene transfer from A. fumigatus?) and at line 68 Shouldn't the organism be Cordyceps farinosa, not A. fumigatus?

> ANSWER : We reivsed it. Please see line 68.

line 32 delete the word other; 

> ANSWER : We deleted it.

line 41 should be in the case

> ANSWER : We changed it. Please see line 39.

line 46 should be fungal not fungi;

> ANSWER : We changed it. Please see line 45.

line 49/50 do the authors mean that only one OSPC gene has been characterised in Aspergillus fumigatus, or that this is the only OSPC gene that has been characterised so far (probably the latter since the following sentence essentially says this)?

> ANSWER : Thank you for this comment. The sentence means latter. We revised the sentence to clarify the confusion. Please see line 49-51.

lines 236-238 Can this sentence be written more clearly? 

> ANSWER : We changed it. Please see line 233-234.

line 266 an, not a

> ANSWER : We changed it. Please see line 274.

Reviewer 3 Report

In this manuscript, Gi Hong An and colleagues wish to improve the knowledge of steroidal triterpenoids in the fungus Cordyceps farinosa. This is an interesting work that can help to better understand biosynthetic pathways and to identify new secondary metabolites, which could be used in particular in the health field. However, this study has been conducted in a confusing way and it is difficult to see clearly the contribution of this study. For instance, why did the authors consider this organism as a model for the study of the triterpenoid pathway? Besides, it is stated in the abstract that “these results will facilitate a greater understanding of regulatory mechanisms”, but in my opinion there is no regulatory aspect addressed here.

This study is composed of two main parts: the functional characterization of a new enzyme (CfaOSC2), and a genomic context study of a gene cluster homologous to genes involved in helvolic acid biosynthesis in A. fumigatus.

I have major comments on these two parts.

The authors detected two OSC genes in C. farinosa: cfaOSC1 and cfaOSC2 that are presented as orthologous to genes coding for lanosterol synthase (Afu5g04080) and protostadienol synthase (AfuOSC3) in A. fumigatus, respectively. Have the enzymes of A. fulmigatus been experimentally characterized, or have the authors simply been guided by their functional annotation? This aspect is not clear in the manuscript.

Why did the authors only consider CfaOSC2 and not also CfaOSC1? According to the manuscript, the yeast strain INVSc1 can synthesize 2,3-oxidosqualene, which could be used as a substrate for both CfaOSC1 (OSLC) and CfaOSC2 (OSPC). The authors should clarify this point.

The authors used retention-time comparison of peaks detected in the presence of CfaOSC2 with peaks in yeast cells co-expressing AfuOSC3/CYP5081A1 to claim that CfaOSC2 produced protosta-13(17),24-dien-3β-ol and protostadienol. These preliminary data can in no way confirm the identity of the molecules. The structural characterization of these compounds requires further investigation, through purification and NMR analysis.

In the synteny comparison chapter, it is not obvious whether there is a relationship between the characterized cfaOSC2 gene in the previous chapter and the one belonging to the gene cluster in synteny with A. fumigatus involved in helvolic acid. This should be explicit in the text. The authors observed that in the C. farinosa cluster, two genes were missing and propose that because of this, the final product is 16 β-acetyloxy-29-norprotosta-1,17(20)Z,24-trien-3-one-21-oic acid, instead of helvolic acid. However, the authors do not mention whether they considered the possibility that the two missing genes are elsewhere in the genome. Alternatively, did the authors consider the existence of non-homologous isofunctional enzymes (NISE)? This part of the study remains speculative. At the very least, some experimental data should be considered. For example, would it be possible to detect (GC/MS) a compound compatible with 16 β-acetyloxy-29-norprotosta-1,17(20)Z,24-trien-3-one-21-oic acid or helvolic acid? This would help to argue this part of the work.

Other comments:

It is stipulated that steroidal triterpenoids are considered as secondary metabolites (line 33). It would be useful to have a Figure linking these metabolic pathways to central metabolism. In this respect, Figure 1 is rather incomplete. For instance, the meaning of solid and dot arrows have to be explained.

It is not easy to understand what type of genetic material the authors used for their heterologous expression system: DNA or cDNA? The mention “cDNA” appears only once (line 65); elsewhere it is said “DNA”. Please explain.

The legend of figure 5 is very incomplete and does not allow understanding the nature of the experiments reported. What is the background of yeast cells in each experiment? “TIC” (Total Ion Current) has to be defined, as well as “EIC” (Extracted Ion Chromatogram). “EIC 189” has to be further explained. What is the nature of the Internal Standard? Figure 5 Part b has to be explained in more details. What is the nature of the 4 panels depicted?

Lines 78-79: these data belong to the Results section.

In conclusion, the manuscript could be suitable for publication but only after it has been revised to address these points

Author Response

Thank you for valuable comments. We did our best to revise this manuscript according to reviewer's comment. Revised parts were remarked by red color. 

Once again, thank you for comments.

Reviwer 3

For instance, why did the authors consider this organism as a model for the study of the triterpenoid pathway?

> ANSWER : C. farinosa is a common fungus having interesting ecological habit (insect-pathogen), and easy to grow on an artificial medium. At the preliminary comparative genome analysis of Hypocrealean species, we recognized that only C. farinosa has gene families which are structurally similar to a well-known gene cluster related to helvolic acid biosynthesis in A. fumigatus. Please see line 64-65.

 Besides, it is stated in the abstract that “these results will facilitate a greater understanding of regulatory mechanisms”, but in my opinion there is no regulatory aspect addressed here.

> ANSWER : We fully agree with your opinion. We revised regulatory mechanisms to biosynthetic mechanisms. Please see line 24 and 303.

The authors detected two OSC genes in C. farinosa: cfaOSC1 and cfaOSC2 that are presented as orthologous to genes coding for lanosterol synthase (Afu5g04080) and protostadienol synthase (AfuOSC3) in A. fumigatus, respectively. Have the enzymes of A. fulmigatus been experimentally characterized, or have the authors simply been guided by their functional annotation? This aspect is not clear in the manuscript.

> ANSWER : We could not get the authentic stardard for two compounds from any chemical company. So we requested Prof. Mitsuguchi to provide the standard. But, instead of the standard, He provide me with a AfuOSC3/CYP5081A1 vector that already was identified by NMR analysis. Their data including GC-MS and NMR analyses was published. Please see ref [6, 7]. Therefore, we made genes in the received vector expressed using our yeast expression system and then used as standard. 

Why did the authors only consider CfaOSC2 and not also CfaOSC1? According to the manuscript, the yeast strain INVSc1 can synthesize 2,3-oxidosqualene, which could be used as a substrate for both CfaOSC1 (OSLC) and CfaOSC2 (OSPC). The authors should clarify this point.

> ANSWER : That's right. Although OSLC genes encoded lanosterol synthase, as well as CfaOSC1, are overexpressed by using yeast expression system, we could not distinguish between endogenous lanosterol and lanosterol from genes as accumulated products in yeast cells. Generally, purified-synthetic protein is used to characterize lanosterol synthase gene. Since a goal of this study is to characterize a gene related to protostane-triterpenoid biosynthesis, we performed only CafOSC2 characterization. As mentioned in Introduction, a lot of OSLC genes already have been reported.

The authors used retention-time comparison of peaks detected in the presence of CfaOSC2 with peaks in yeast cells co-expressing AfuOSC3/CYP5081A1 to claim that CfaOSC2 produced protosta-13(17),24-dien-3β-ol and protostadienol. These preliminary data can in no way confirm the identity of the molecules. The structural characterization of these compounds requires further investigation, through purification and NMR analysis.

> ANSWER : As mentioned by above, we could not get the authentic stardard for two compounds from any chemical company. So we requested Prof. Mitsuguchi to provide the standard. But, instead of the standard, He provide me with a AfuOSC3/CYP5081A1 vector that already was identified by NMR analysis. Their data including GC-MS and NMR analyses was published. Please see ref [6, 7]. Therefore, we made genes in the received vector expressed using our yeast expression system and then used as standard. We think that there is no doubt about products from AfuOSC3/CYP5081A1 genes as the standard.

In the synteny comparison chapter, it is not obvious whether there is a relationship between the characterized cfaOSC2 gene in the previous chapter and the one belonging to the gene cluster in synteny with A. fumigatus involved in helvolic acid. This should be explicit in the text. 

> ANSWER : We changed OSC into OSPC to clarify the confusion. Please see Figure 6. And additional information about genes related to helvolic acid biosythesis were added in line 267-273.

The authors observed that in the C. farinosa cluster, two genes were missing and propose that because of this, the final product is 16 β-acetyloxy-29-norprotosta-1,17(20)Z,24-trien-3-one-21-oic acid, instead of helvolic acid. However, the authors do not mention whether they considered the possibility that the two missing genes are elsewhere in the genome. Alternatively, did the authors consider the existence of non-homologous isofunctional enzymes (NISE)? This part of the study remains speculative. 

> ANSWER : We tried to detect the two missing genes by using the assembled sequences, but did not catch them even applying loose parameters. As mentioned by a reviwer, it is possible that non-homologous isofunctional enzymes exist somewhere. However, unfortunately, preceding research data on NISE is limited in fungal groups.

At the very least, some experimental data should be considered. For example, would it be possible to detect (GC/MS) a compound compatible with 16 β-acetyloxy-29-norprotosta-1,17(20)Z,24-trien-3-one-21-oic acid or helvolic acid? This would help to argue this part of the work.

> ANSWER : Thank you for this comment. We fully agree with your opinion. Unfortunately, we did not dectect any peak for 16 β-acetyloxy-29-norprotosta-1,17(20)Z,24-trien-3-one-21-oic (ANToic) and helvolic acid in C. farinosa mycelium by HPLC analysis. 

We don't know the reason why did not detect ANToic in C. farinosa, because it is no any information whether or not the compound is accumulated in C. farinosa mycelium. If ANToic is used for next any compound as a precursor, it may be not accumulated in C. farinosa.

Therefore, because we don't know existence of ANToic in C. farinosa, we have no choice but to infer ANToic as a final product, considering the gene cluster. To obviously find out whether ANToic was synthesized and accumulated in C. farinosa, future studies for all gene characterization by heterologus expression system are needed as mentioned in the text. The other hand, to suggest an evidence for no detection of helvolic acid caused by two gene deletion, we added Supplementary Figure S1 including the data that helvolic acid was not dectected in C. farinosa. Please see line 275-278 and the end page.

Other comments:

It is stipulated that steroidal triterpenoids are considered as secondary metabolites (line 33). It would be useful to have a Figure linking these metabolic pathways to central metabolism. In this respect, Figure 1 is rather incomplete. For instance, the meaning of solid and dot arrows have to be explained.

> ANSWER : Generally, solid arrow means one step or definite step. Dot arrow means multiple steps or indefinite steps. We added it. Please see line 52.

It is not easy to understand what type of genetic material the authors used for their heterologous expression system: DNA or cDNA? The mention “cDNA” appears only once (line 65); elsewhere it is said “DNA”. Please explain.

> ANSWER : We change DNA into cDNA to clarify cDNA synthesized by reverse transcripts. Please see line 199, 230, 232.

The legend of figure 5 is very incomplete and does not allow understanding the nature of the experiments reported. What is the background of yeast cells in each experiment? “TIC” (Total Ion Current) has to be defined, as well as “EIC” (Extracted Ion Chromatogram). “EIC 189” has to be further explained. What is the nature of the Internal Standard? Figure 5 Part b has to be explained in more details. What is the nature of the 4 panels depicted?

> ANSWER : According to reviwer's comment, Fig 6 was revised.

Lines 78-79: these data belong to the Results section.

> ANSWER : According to reviwer's comment, these data have been shifted to the Resutls section. Please see line 177-187.

Round 2

Reviewer 1 Report

Authors have addressed the comments satisfactorily. Although there are minor text edits that are needed.

Minor comment:

1) In Line 64, authors might want to start the sentence with "We chose C. farniosa because…." rather than "Because"

2) Line 181 - similarity instead of similar

3) Mention gene details when the gene is first mentioned in main text (e.g. AfuOSC3 in line 157)

Author Response

Thank you for valuable comments. We have revised all you point-out as below and you can check it in manuscript. Thanks a lot.

Minor comment:

1) In Line 64, authors might want to start the sentence with "We chose C. farniosa because…." rather than "Because"

> ANSWER : Thank you for this comment. We revised it. Please see line 64.

2) Line 181 - similarity instead of similar

> ANSWER : Thank you for this comment. We revised it. Please see line 182.

3) Mention gene details when the gene is first mentioned in main text (e.g. AfuOSC3 in line 157)

> ANSWER : Thank you for this comment. I don't know why that was not revised last time. Sorry about that. We revised it. Please see line 158 and 199.

Reviewer 3 Report

I thank the authors for their feedback. However, some major aspects are not yet addressed in a satisfactory manner.

Remark Report 1: The authors used retention-time comparison of peaks detected in the presence of CfaOSC2 with peaks in yeast cells co-expressing AfuOSC3/CYP5081A1 to claim that CfaOSC2 produced protosta-13(17),24-dien-3β-ol and protostadienol. These preliminary data can in no way confirm the identity of the molecules. The structural characterization of these compounds requires further investigation, through purification and NMR analysis.

> ANSWER : As mentioned by above, we could not get the authentic stardard for two compounds from any chemical company. So we requested Prof. Mitsuguchi to provide the standard. But, instead of the standard, He provide me with a AfuOSC3/CYP5081A1 vector that already was identified by NMR analysis. Their data including GC-MS and NMR analyses was published. Please see ref [6, 7]. Therefore, we made genes in the received vector expressed using our yeast expression system and then used as standard. We think that there is no doubt about products from AfuOSC3/CYP5081A1 genes as the standard.

I have no doubt that protosta-13(17),24-dien-3β-ol and protostadienol were correctly characterized  by Mistuguchi et al. But here, a simple comparison of retention times, without any structural information, cannot be considered as proof of identity. Furthermore, a vector for gene expression in yeast cannot be considered as a standard. The authors must change this designation. The authors should also modify the text in the sense of suggesting that the two products could be protosta-13(17),24-dien-3β-ol and protostadienol, and that further studies are needed to confirm their identity.

Remark Report 1: At the very least, some experimental data should be considered. For example, would it be possible to detect (GC/MS) a compound compatible with 16 β-acetyloxy-29-norprotosta-1,17(20)Z,24-trien-3-one-21-oic acid or helvolic acid? This would help to argue this part of the work.

> ANSWER : Thank you for this comment. We fully agree with your opinion. Unfortunately, we did not dectect any peak for 16 β-acetyloxy-29-norprotosta-1,17(20)Z,24-trien-3-one-21-oic (ANToic) and helvolic acid in C. farinosa mycelium by HPLC analysis.

We don't know the reason why did not detect ANToic in C. farinosa, because it is no any information whether or not the compound is accumulated in C. farinosa mycelium. If ANToic is used for next any compound as a precursor, it may be not accumulated in C. farinosa.

Therefore, because we don't know existence of ANToic in C. farinosa, we have no choice but to infer ANToic as a final product, considering the gene cluster. To obviously find out whether ANToic was synthesized and accumulated in C. farinosa, future studies for all gene characterization by heterologus expression system are needed as mentioned in the text. The other hand, to suggest an evidence for no detection of helvolic acid caused by two gene deletion, we added Supplementary Figure S1 including the data that helvolic acid was not dectected in C. farinosa. Please see line 275-278 and the end page.

The authors finally do not have evidence of the function of the gene cluster, since neither the final product helvolic acid nor its precursor ANToic acid could be detected. In conclusion, this chapter of the study remains speculative. The authors may try recombinant gene expression in yeast (for example) to get substantial evidence of gene function. In the absence of additional results, this study brings little new information. The Figure S1 is not useful.

Remark Report 1: The legend of figure 5 is very incomplete and does not allow understanding the nature of the experiments reported. What is the background of yeast cells in each experiment? “TIC” (Total Ion Current) has to be defined, as well as “EIC” (Extracted Ion Chromatogram). “EIC 189” has to be further explained. What is the nature of the Internal Standard? Figure 5 Part b has to be explained in more details. What is the nature of the 4 panels depicted?

> ANSWER : According to reviwer's comment, Fig 6 was revised.

The legend remains incomplete and Figure 5 still cannot be understood. In Figure 5 (a), the ions of m/z = 189 are not explained. Furthermore, I could not find mention of these molecules in the text. The nature of the internal standard is still not described. The remark in report 1 is still valid. What is the information given by Figure 5 (b)? To be more specific, I don't see how Figure 5 supports the text and how it supports the identity of the studied compounds. The authors would probably gain a lot of clarity by thoroughly describing Figure 5 and the related experiments.

Other comments:

Lines 12 and 275: please modify “Cordyceps farinose”.

Line 241: please modify “Mitsuguch et al.”

Line 272: “highly identical”. Please modify. Two sequences are identical or are not identical.

Author Response

Thank you for valuable comments. We have attached a file including answers for your comments. Please see it.
